# High-resolution X-ray imaging via spatially decoupled heavy-atom antennas in organic scintillators

Chensen Li [1,8] ✉, Yaohui Li[2,8], Minghui Wu[3], Fan-Cheng Kong[4], Binxia Jia[2], Zonghang Liu[5], Xilong Wei[5], Philip C. Y. Chow [4], Zhicheng Wang[6], Xiaoming Li [6], Bo Xu [1] ✉, Zheng Zhao [5], Ryan T. K. Kwok [7], Jacky W. Y. Lam [7] ✉, Yucheng Liu [2] ✉, Shengzhong Frank Liu[2] & Ben Zhong Tang [5] ✉

Organic scintillators are promising for X-ray imaging due to low cost, sustainability, and tunable structures, but their commercial use is limited by poor understanding of charge transfer design for balancing light yield, decay, and bandwidth. Here, we propose a spatially decoupled heavy atom antenna strategy, integrating alkyl bromides into a hybridized local and charge-transfer scaffold to create a scintillator. This architecture leverages the moderate charge-transfer state to deliver an optimal combination of a short radiative lifetime (3.74 ns), a narrow radioluminescence bandwidth (56 nm), a large Stokes shift (110 nm) and a high photoluminescence quantum yield of 100%. As a result, this scintillator exhibits excellent radioluminescence properties, rendering it suitable for highly sensitive X-ray detections. In this work, we elucidate a general design principle for creating high-performance scintillators that meet the stringent multi-property demands of advanced X-ray imaging applications.

X-ray imaging systems transform X-ray photons passing through the target object into observable visual images for the assessment of internal structural features, and such systems have found widespread applications in industrial nondestructive testing, high-energy physics, medical radiography, security inspection, and astronomical exploration[1–4]. Spatial resolution is one of the most important indexes for the practical application of X-ray imaging, which directly leads to the quality of imaging[5]. Currently, the spatial resolution remains unsatisfactory for fine X-ray imaging applications—such as high-resolution X-ray imagers used in nondestructive testing within the microelectronic industry—which requires a scintillator with sufficiently high spatial resolution to distinguish the delicate structural details of electronic chips[6]. Organic scintillators exhibit inherent advantages, such as abundant resources, high mechanical flexibility,

[1]Key Laboratory for Soft Chemistry and Functional Materials of Ministry of Education, School of Chemistry and Chemical Engineering, Nanjing University of Science and Technology, Nanjing, Jiangsu, China. [2]School of Materials Science and Engineering, Shaanxi Normal University, Xi'an, China. [3]Joint International Research Laboratory of Animal Health and Food Safety of Ministry of Education & Single Molecule Nanobiology Laboratory (Sinmolab), Nanjing Agricultural University, Nanjing, Jiangsu, China. [4]Department of Mechanical Engineering, The University of Hong Kong, Pokfulam, Hong Kong, China. [5]Guangdong Basic Research Center of Excellence for Aggregate Science, School of Science and Engineering, Shenzhen Institute of Aggregate Science and Technology, The Chinese University of Hong Kong, Shenzhen, Guangdong, China. [6]MIIT Key Laboratory of Advanced Display Materials and Devices, Institute of Optoelectronics & Nanomaterials, College of Materials Science and Engineering, Nanjing University of Science and Technology, Nanjing, Jiangsu, China. [7]Department of Chemistry and Hong Kong Branch of Chinese National Engineering Research Center for Tissue Restoration and Reconstruction, The Hong Kong University of Science and Technology, Kowloon, Hong Kong, China. [8]These authors contributed equally: Chensen Li, Yaohui Li. ✉e-mail: chensenli@njust.edu.cn; boxu@njust.edu.cn; chjacky@ust.hk; liuyc@snnu.edu.cn; tangbenz@cuhk.edu.cn

easy processing, low cost and large-area fabrication, and have received much attention in recent years[7–12]. To develop efficient organic scintillators, some key parameters are required to be optimized, including (i) the short radiative lifetime (<10 ns) enables quickly dynamic X-ray imaging and medical diagnosis[10]; (ii) the narrow full-width at half-maximum (FWHM) (<60 nm) radioluminescence (RL) can reduce scattering and noise, enhance resolution and contrast, reduce crosstalk, and improve detection clarity and accuracy of imaging[13]; (iii) the large Stokes shift (>100 nm) reduces self-absorption of X-ray sources and ensures the full utilization of excitation energy[8,9]; and (iv) the high photoluminescence quantum efficiency (PLQY) (>90%) is closely related to the high conversion efficiency of scintillators from X-ray to luminescence[14].

Charge transfer (CT)[15] processes play a pivotal role in regulating key photophysical properties of organic luminescent materials, including radiative lifetime, emission bandwidth, Stokes shift, and luminescence efficiency—parameters critical for optimizing the performance of organic scintillators. While current organic scintillators have made significant progress in improving triplet exciton utilization efficiency, they still cannot realize molecular design by regulating photophysical properties via CT modulation. This limitation leaves them unable to meet the requirements of high-resolution X-ray imaging (Fig. 1a)[7–10]. For instance, room-temperature phosphorescence (RTP)[16] scintillators exhibit narrow emission spectra owing to weak CT but possess long phosphorescence lifetimes (-ms)[7]. Thermally activated delayed fluorescence (TADF)[17,18] scintillators, characterized by strong CT, feature broad emission spectra and long delayed fluorescence lifetimes (-μs)[8,9]. While hot exciton[19] scintillators feature short fluorescence lifetimes (-ns), their lack of donor-acceptor units renders them unable to form intramolecular CT, making it challenging to systematically tune their photophysical properties[10]. Accordingly, it is imperative to propose innovative molecular design strategies for the holistic regulation of diverse photophysical properties of molecules through modulating CT degree, thereby enabling highly efficient RL. Hybridized local and charge transfer (HLCT)[20,21] molecules with partially twisted donor-acceptor architectures exhibit a moderate degree of CT, enabling simultaneous fulfillment of diverse photophysical requirements for efficient organic scintillators. Specifically, HLCT scintillators exhibit two compatible characteristics: a locally-excited (LE) state with large oscillator strength and a CT state with enhanced intersystem crossing (ISC) ability (Fig. 1b). The former contributes to short radiative lifetimes and narrow emission spectra, while the latter enables high triplet exciton utilization via high-lying reversed ISC (hRISC) from upper triplet (T_n) to the lowest singlet (S_1) states. In addition, weakly-bound CT excitons afford a large Stokes shift. Therefore, these advantages enable HLCT scintillators to hold great potential in high-resolution X-ray imaging.

Heavy atoms play an essential role in organic scintillators, as their high atomic number (Z) enables efficient absorption of X-ray photons through the photoelectric effect[22]. To enhance X-ray absorption cross-sections, these organic scintillators typically involve the direct covalent attachment of halogen heavy atoms (Cl, Br, and I) to peripheral benzene rings[9,10]. However, the strong conjugated heavy atom-π interactions enable hole-electron delocalization onto the halogen atom, fostering pronounced nonradiative transitions—likely driven by enhanced vibronic coupling and spin-orbit-induced energy dissipation pathways[23,24]. This intrinsic trade-off—where improved X-ray absorption is accompanied by degraded radiative efficiency—severely restricts the RL performance of organic scintillators, limiting their practical application in high-sensitivity imaging scenarios. Therefore, innovative heavy-atom engineering is essential to address this long-standing dilemma by aiming to decouple the dual effects of heavy atoms: retaining their high X-ray absorption capacity while substantially suppressing the detrimental impact of non-radiative transitions on RL performance, which enables synergistic optimization of

the absorption-efficiency dual core metrics for developing high-performance organic scintillators. The antenna effect[25] in coordination chemistry is a sensitization mechanism where organic ligands act as molecular antennas, absorbing light and funneling energy nonradiatively to a central metal ion (e.g., lanthanide) (Fig. 1c). This overcomes the ion's intrinsically weak light absorption, enabling enhanced metal-centered luminescence. Inspired by the antenna effect, we speculate that heavy atoms can act as X-ray-absorbing antenna units to transfer energy to HLCT molecules (Fig. 1c). These RL centers not only efficiently accept this energy for RL emission but also suppress nonradiative loss from direct heavy-atom interactions in traditional scintillators. This suppression stems from minimized hole/electron distributions on bromines and a reduced spin-orbit coupling constant between lowest triplet (T_1) and ground state (S_0) (SOC(T_1,S_0)), thus boosting RL efficiency and ultimately providing a feasible route to high-efficiency X-ray scintillators.

In this work, a large Stokes shift (110 nm), high PLQY (100%) and short decay time (3.74 ns) of organic scintillators BTD-HeBr are achieved simultaneously by HLCT emitters with attaching through-space bromines. It exhibits a narrow RL spectrum with FWHM of 56 nm, a low detection limit of 84.6 nGy s$^{-1}$ and a high X-ray imaging resolution. This finding provides a powerful design approach and promising alternative materials for fabricating organic X-ray scintillators with excellent sensitivity, high resolution, and favorable stability.

## Results
### Theoretical calculations
To validate our hypothesis, we designed two HLCT scintillators based on donor (9,9-dimethyl-9H-fluoren (FL)) and acceptor (benzothiadiazole (BTD)) with a moderate dihedral angle (-36°) (Fig. 1)[26]. Introducing conjugated bromine atoms to improve SOC and X-ray absorption cross-section, resulting in BTD-FLBr. To minimized non-radiative decay, the nonconjugated alkyl bromides were introduced to obtain BTD-HeBr. They can be easily synthesized by one-pot method with a high reaction yield under the same reaction conditions by using different substituted reactants (Supplementary Figs. 1–7). Theoretical calculations are conducted by exploiting the density functional theory (DFT) and time-dependent (TD) DFT (PBE0/6-31g(d,p))[27,28]. The scintillators exhibit partial highest occupied molecular orbital and lowest unoccupied molecular orbital separations, as well as distinct HLCT characters, with a moderate intramolecular CT character expected (Supplementary Fig. 8). Natural transition orbital (NTO) analyzes[29,30] are also performed to explore the excited states and transition characters (Fig. 2a). The hole of S_1 was mostly distributed on the whole conjugated system, while the electron was mainly located on the BTD acceptor. Therefore, the S_1 presents a hybrid feature of LE (42.3% and 43.8%) and CT (57.7% and 57.2%) for BTD-FLBr and BTD-HeBr, respectively (Supplementary Fig. 9). Specifically, in BTD, the electron composition was significantly higher at -86% but the hole composition was -30%. This unit results in a -49% overlap and demonstrates a typical HLCT feature. In contrast, the fluorene exhibits a main hole composition of -71%, an electron composition of -14%, and an overlap of -31%, which can be considered to show CT feature (Supplementary Figs. 10 and 11). These results illustrate that the HLCT features of the two scintillators mainly come from BTD acceptors. The HLCT characteristics are beneficial for the luminescence transition and exhibit short radiative lifetimes. Specifically, the LE character of ¹HLCT state of the two HLCT scintillators renders a high oscillator strength ($f = 0.679$ and 0.554 for BTD-FLBr and BTD-HeBr, respectively) from S_1 to S_0 (Fig. 2b).

In further calculations, the energy-level arrangements of BTD-FLBr and BTD-HeBr are studied (Fig. 2b). The excited state levels of the two HLCT scintillators exhibit a tiny energy difference (0.02−0.03 eV) between S_1 and T_2. A small energy difference (0.06−0.11 eV) between S_1 and T_3 was also found, which implies that the spin-flip routes from

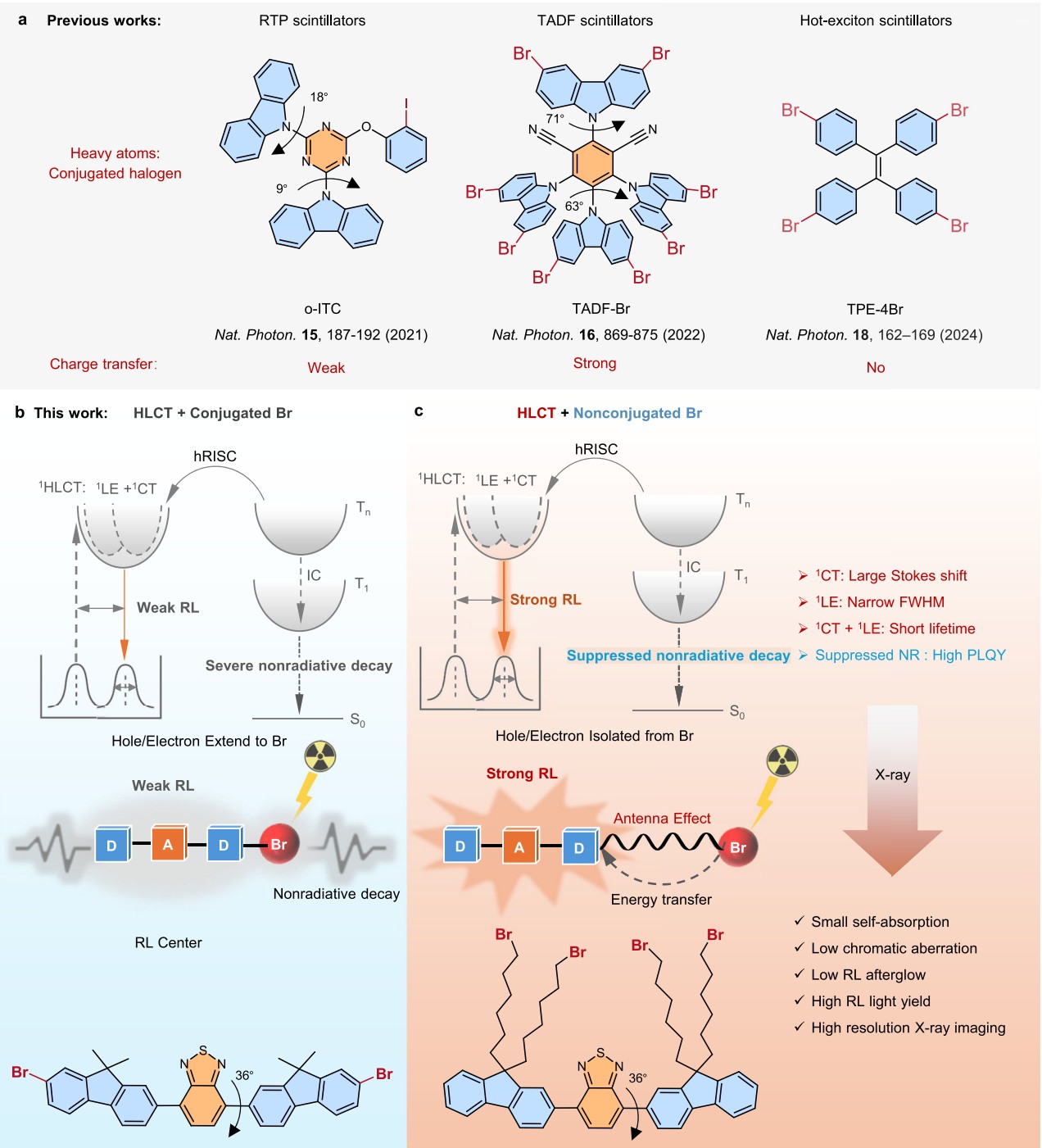

**Fig. 1 | Illustration of the spatial decoupled heavy atom-π strategy in hybridized local and charge transfer (HLCT) scintillators on the enhancement of radioluminescence (RL) performances. a** The three types of organic scintillators with similar conjugated heavy atoms strategy and different charge transfer characters. **b** The conjugated coupled bromine-π interactions are present in the HLCT scintillators, where hole/electron extend to bromines, inducing high SOC(T$_1$,S$_0$) and severe non-radiative transitions that weaken RL efficiency. **c** The spatially decoupled bromine-π interactions are present in the HLCT scintillators, where hole/electron are isolated from bromines, inducing low SOC(T$_1$,S$_0$) and suppressed non-radiative decays that enhance RL efficiency. The HLCT scintillators with moderate CT characters and their advantages of short radiative lifetime, large Stokes shift, and narrow full width at half maximum (FWHM). RTP, room temperature phosphorescence; TADF, thermally activated delayed fluorescence; $^1$CT, charge transfer singlet state; $^1$LE, localized singlet sate; $^1$HLCT, hybridized local and charge transfer singlet state; hRISC, high-lying reversed intersystem crossing; T$_1$, the first triplet state; T$_n$, the higher triplet state; IC, internal conversion; NR, non-radiative transition; D, donor; A, acceptor.

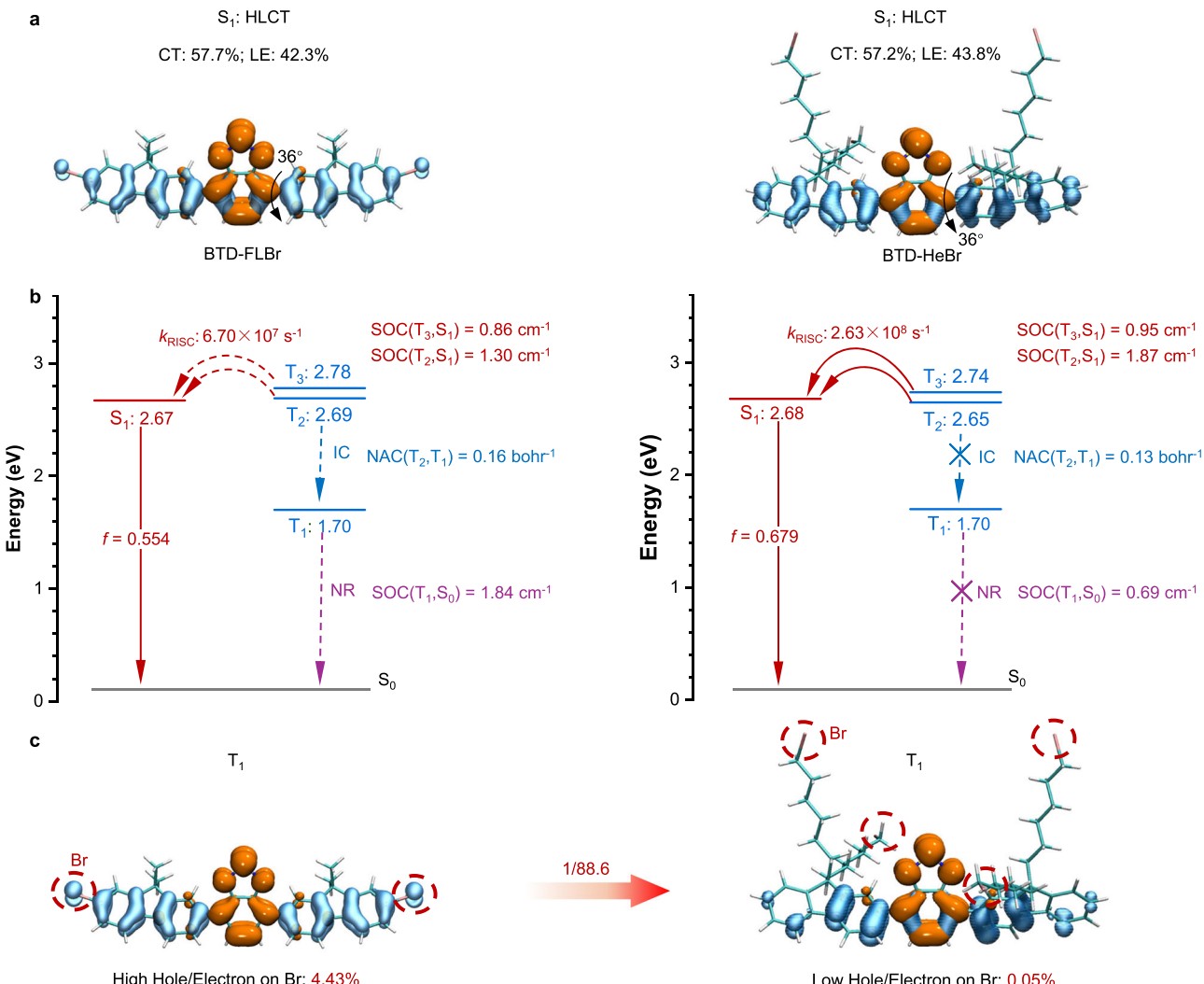

**Fig. 2 | Theoretical calculations of the excited states for the coupled and decoupled Br-π HLCT scintillators. a** The hole and electron distributions of $S_1$ of BTD-FLBr and BTD-HeBr. **b** Diagram of energy levels and properties of the excited states and the photophysical processes of BTD-FLBr and BTD-HeBr, including their RISC rates from triplet to singlet, SOC constants between $T_{2,3}$ and $S_1$, NAC constants between $T_2$ and $T_1$, and SOC constants between $T_1$ and $S_0$. **c** The hole and electron distributions of $T_1$ of BTD-FLBr and BTD-HeBr. $k_{RISC}$, rate constant of the RISC process; IC, interconversion; NAC, vibronic nonadiabatic coupling; SOC, spin-orbital coupling.

higher triplet ($T_{2,3}$) to $S_1$ states are feasible. In addition, large SOC matrix elements are critical for enhancing the RISC rate[31] (Fig. 2b and Supplementary 12,13 and Eq. 1). BTD-HeBr shows a significant enhancement in SOC values compared to BTD-FLBr: its SOC ($T_2$, $S_1$) increases from BTD-FLBr's 1.30 cm$^{-1}$ to 1.87 cm$^{-1}$, while SOC ($T_3$, $S_1$) rises from 0.86 cm$^{-1}$ to 0.95 cm$^{-1}$. This marked improvement in SOC interactions strengthens the dual RISC channels ($T_2 \rightarrow S_1$ and $T_3 \rightarrow S_1$) more effectively, leading to a dramatic boost in the RISC rate: BTD-HeBr's $k_{RISC}$ reaches $2.63 \times 10^8$ s$^{-1}$, which is nearly four times higher than BTD-FLBr's $6.70 \times 10^7$ s$^{-1}$. These results highlight that the decoupling strategy for Br-π interactions does not diminish the RISC process in BTD-HeBr. On the contrary, the abundant through-space Br-π interactions introduced by this decoupling design amplified the SOC, thereby driving a substantial increase in the RISC rate compared to BTD-FLBr (Supplementary Figs. 14 and 15). This confirms that rational regulation of spatial Br-π interactions via decoupling engineering is key to enhancing SOC-mediated RISC efficiency, laying the foundation for BTD-HeBr's superior X-ray scintillation performance.

Apart from an efficient RISC process, a low nonradiative decay of HLCT emitters is also a significant factor for achieving efficient X-ray

detection. Notably, a low energy level of $T_1$ (1.70 eV) and a large energy gap (0.95−0.99 eV) between $T_1$ and $T_2$ are observed, which can effectively suppress interconversion (IC) transition between them. As illustrated in Fig. 2b, the vibronic nonadiabatic coupling (NAC) matrix element between $T_1$ and $T_2$ states can be calculated to further evaluate the possibility of the existence of the IC process. The $k_{IC}$ ($T_2 \rightarrow T_1$) was calculated by using the following equation[32]:

$$k_{IC}(T_2 \rightarrow T_1) \propto \left| \frac{\langle NAC(T_2, T_1) \rangle}{E(T_2) - E(T_1)} \right|^2 \quad (1)$$

where $\langle NAC(T_2, T_1) \rangle$ is the NAC matrix between $T_2$ and $T_1$, $E(T_2)/E(T_1)$ is the energy level of $T_2/T_1$. The calculated root-mean-square NACs are 0.16 and 0.13 bohr$^{-1}$ for BTD-FLBr and BTD-HeBr, respectively. In contrast to BTD-FLBr, BTD-HeBr exhibits a smaller NAC ($T_2$, $T_1$). This reduction in NAC arises from the decoupled Br-π interactions in BTD-HeBr, which alleviates the orbital overlap between triplet states ($T_2$ and $T_1$) compared to the coupled Br-π configuration in BTD-FLBr (Supplementary Figs. 16 and 17). As a result, the decoupled Br-π design in BTD-HeBr reduces nonradiative energy dissipation via IC

with suppressed $k_{IC}$ ($T_2 \rightarrow T_1$), thereby favoring efficient RL performance.

Furthermore, the non-radiative transition from $T_1$ to $S_0$ is a more important deactivation route of the triplet exciton. The $k_{nr}(T_1 \rightarrow S_0)$ was calculated using the following equation[33]:

$$k_{nr}(T_1 \rightarrow S_0) \propto \frac{\langle SOC(T_1, S_0) \rangle^2}{\{E(T_1) - E(S_0)\}^2 + \gamma^2} \qquad (2)$$

where $\langle SOC(T_1, S_0) \rangle$ is the SOC matrix between $T_1$ and $S_0$, $\gamma$ is the broadening of the line-shape function for ISC (Fig. 2b). The SOC ($T_1$, $S_0$) in BTD-FLBr decreases significantly from 1.84 to 0.69 cm$^{-1}$ in BTD-HeBr, indicating spatially decoupled heavy atom strategy in BTD-HeBr significantly suppressed $k_{nr}(T_1 \rightarrow S_0)$. The decrease of SOC ($T_1$, $S_0$) in BTD-HeBr can attributed to the decrease of contribution of bromine to the excited state wave function. For BTD-FLBr, the contribution of bromine atoms to hole/electron distribution in $T_1$ is as high as 4.43%, enhancing vibronic coupling and spin-orbit-induced energy dissipation pathways. (Fig. 2c, and Supplementary Fig. 17). In contrast, the proportion of bromine's electron/hole distribution in the $T_1$ state of BTD-HeBr is as low as 0.05%, which mitigates orbital hybridization between heavy atoms and π orbitals. Bromine's reduced participation in the $T_1$ wavefunction diminishes the probability of energy transfer to low-frequency molecular vibrations or rotations, thereby weakening the vibronic coupling between electronic states and vibrational modes. These results indicate that bromine's contribution to the excited-state wavefunction is greatly reduced through spatial decoupling between π systems and bromine atoms, resulting in markedly low nonradiative decay and enabling the achievement of efficient organic X-ray scintillators.

## Photophysical properties

The two HLCT scintillators showed similar absorption bands in the ultraviolet and blue spectral ranges in dilute toluene solutions. A broad, featureless absorption band was observed at around 350–500 nm with peak at 410~413 nm, which is assigned to the intra-molecular charge transfer absorption associated with the electron transfer from the fluorene to benzothiadiazole (Fig. 3a). Noteworthily, the HLCT scintillators with moderate charge transfer, while exhibiting high absorption intensity, retain high radiative efficiency, which is consistent with the high oscillator strength as mentioned above. In contrast, the steady-state photoluminescence (PL) spectra of these HLCT scintillators show an unstructured emission band from 480 to 640 nm centered at ~520 nm. Owing to the CT character of the HLCT state, absorption-emission spectral overlaps are small—only 2.9% for BTD-FLBr and 1.5% for BTD-HeBr, respectively. This gives rise to a large Stokes shift (~110 nm) with a near-free reabsorption feature, a pre-requisite for high-performance scintillators.

The solvatochromic test results to verify their HLCT characters are shown in Fig. 3b. When increasing solvent polarity, the UV-vis absorption spectra (Supplementary Fig. 18) of the HLCT scintillators remain virtually unchanged, indicating minimal solvent dependence of the ground-state dipole moment. In PL spectra of low-polar solvents, the PL peaks are nearly unchanged, indicating a LE-like singlet character. In high polar solvents, their PL spectra (Supplementary Fig. 19) show a ~40 nm red-shift accompanied by spectral broadening with increasing solvent polarity, indicating excited states with certain CT components. Therefore, the two HLCT scintillators contain both the intrinsic LE and CT excited states and demonstrates HLCT characteristics. In addition, Stokes shifts ($v_a - v_f$) calculated from absorption/emission spectra were plotted against solvent polarity ($f$) using the Lippert–Mataga model to characterize solvent-emitter dipole interactions (Fig. 3b and Supplementary Table 1). For HLCT scintillators, these plots show a two-segment line corresponding to two distinct exciton states with distinct small/large dipole moments

in low-polar ($f \leq 0.15$) and high-polar ($f \geq 0.15$) solvents, respectively. In the high-polarity region, BTD-FLBr and BTD-HeBr exhibit excited-state dipole moment ($\mu_e$) of 16.8 and 18.4 D, respectively—values close to that of the typical CT molecule 4-($N,N$-dimethylamino)ben-zonitrile ($\mu_e = 23$ D)[34]—indicating CT-dominated character. In the low-polarity region, their $\mu_e$ values (5.7 and 7.30 D, respectively) are comparable to common LE fluorophores like anthracene (4.0–6.0 D)[35]. This confirms unequal LE-CT hybridization in HLCT scintillators, forming LE-based HLCT in low-polar solvents and CT-dominated HLCT in high-polar solvents.

To understand in depth the photophysical properties, we studied the transient spectroscopy (TA) spectra and dynamic trace of BTD-FLBr and BTD-HeBr in tetrahydrofuran, as shown in Fig. 3c. The positive bands from 500 to 600 nm correspond to stimulated emission (SE), while the negative bands from 650 to 730 nm were attributed to the excited-state absorption. The proposed mechanism of transient component evolution is shown in Supplementary Fig. 20. The initial distinct vibronic SE band exhibits structured double peaks at ~540 and ~570 nm, associated with the LE-dominated HLCT state. Later, it blue-shifts to ~540 nm and becomes structureless due to the vibrational cooling coupled planarization process[36], corresponding to the CT-dominated HLCT state. It is inferred that a quick equilibrium exists between the LE and CT parts of the HLCT state. Over longer time delays, moderate spectral red shift to ~560 nm with a monotonous decay in intensity was observed, which implies that LE and CT states undergo interstate coupling forming a HLCT emissive state.

The energy levels of BTD-FLBr and BTD-HeBr can be characterized using their prompt and delayed PL spectra (delayed time: 1 ms) at low temperature (77 K) (Fig. 3d). At 77 K, the prompt PL spectra are similar to those at room temperature but show a slightly blue-shifted peak at 507 nm, which is assigned to $S_1$. In contrast, the delayed PL peak exhibits a significant red shift to ~685 nm, corresponding to $T_1$. Notably, a weak emission band with two peaks at ~515 and ~495 nm is also observed, attributed to the higher-lying triplet states $T_2$ and $T_3$, respectively. The small energy gaps between $T_2/T_3$ and $S_1$ indicate that a fast RISC process is expected in these two HLCT scintillators. These low-temperature PL spectra are consistent with the theoretically calculated energy levels, thereby validating the predicted energy level distributions.

The photophysical properties of the emitters in the aggregate state are crucial for high-sensitive X-ray detection and imaging applications. The curves of PL intensity in water/THF mixtures with different water fractions proves that the two HLCT molecules show aggregation-induced emission characteristics[37–39] (Supplementary Fig. 21), which can demonstrate potential performance as X-ray scintillators. The advancement of scintillators featuring elevated luminous efficiency and superior timing performance constitutes a primary goal in scintillator research. The key to achieving intrinsic efficient and fast organic scintillators is to eliminate the occupation of $T_1$ states without any loss of the utilization efficiency of triplet excitons. Due to the coupled Br-π interaction-induced triplet quenching, the BTD-FLBr exhibits a low PLQY of 52% in neat film (Fig. 3e, and Supplementary Figs. 22 and 23), which is disadvantageous to achieve highly efficient X-ray detection. However, the nonconjugated alkyl chains with decoupled bromines in BTD-HeBr not only enhance the (R)ISC processes but also significantly suppress the non-radiative decays of the triplet states. As a result, the PLQY of BTD-HeBr is largely increased to 100%, implying a promising organic scintillation performance. Additionally, transient PL decay spectra of BTD-FLBr and BTD-HeBr show single exponential decays within nanosecond range of 3-5 ns (Supplementary Fig. 24), signifying that the excited-state responsible for the PL emission originates from the hybridization between LE and CT excited states (HLCT state), not a simple mix-up of the two states. Specifically, the fluorescence lifetime decreases from 4.48 ns for BTD-FLBr to 3.74 ns for BTD-HeBr (Fig. 3f), showing that through-space

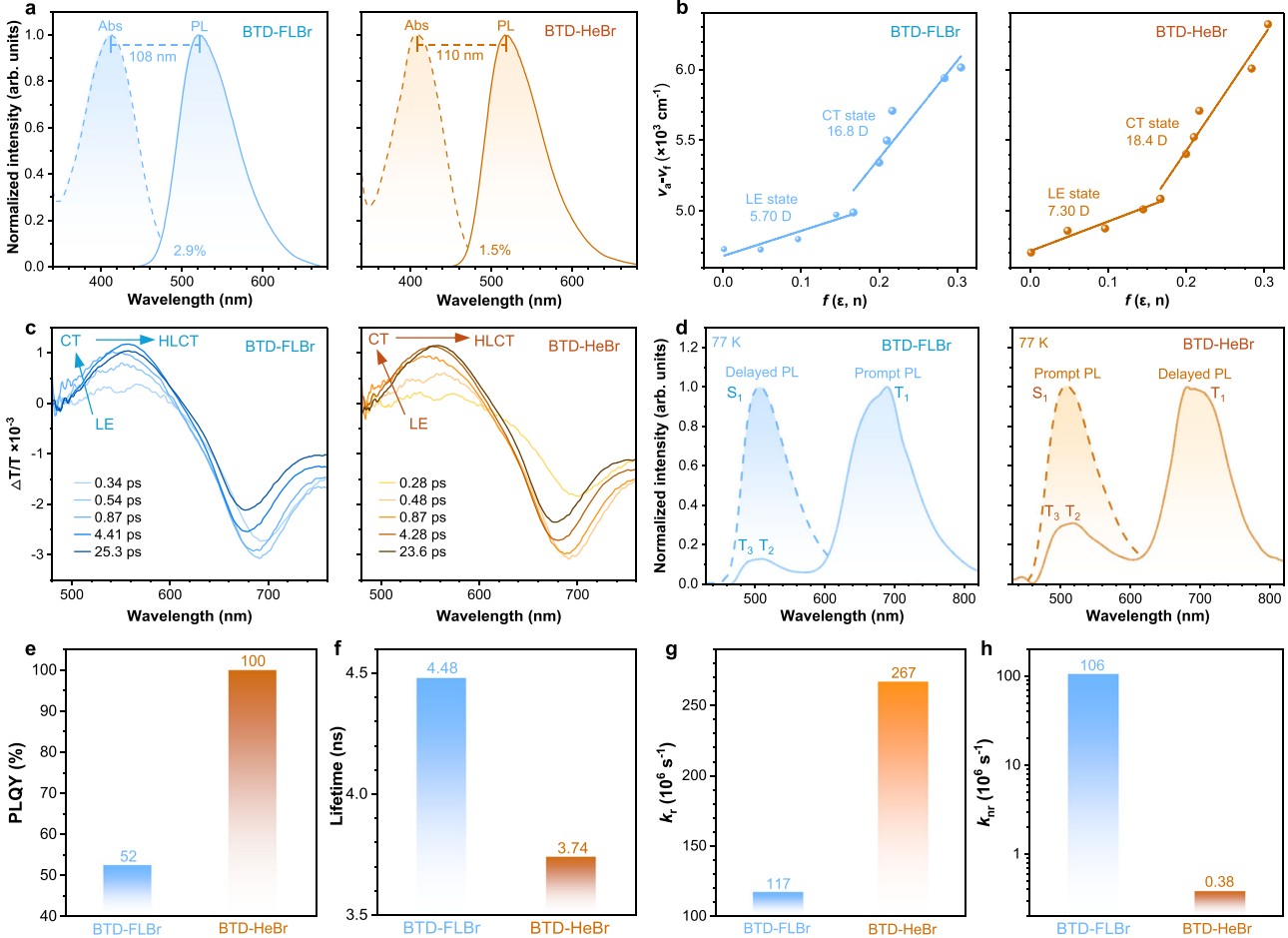

**Fig. 3 | Photophysical properties of the coupled and decoupled Br-π HLCT scintillators. a** UV-vis absorption and photoluminescence (PL) spectra of BTD-FLBr and BTD-HeBr in toluene solutions ($10^{-5}$ M). **b** Linear correlation of orientation polarization ($f$) of solvent media with the Stokes shift ($v_a - v_f$) of them. **c** Time-resolved transient absorption spectra of them in tetrahydrofuran solutions. **d** Prompted PL spectra and delayed PL spectra (delayed time: 1 ms) of them at 77 K. **e** PLQYs of them in neat films. **f** PL decay lifetimes of them. **g** Radiative transition rate ($k_r$) of them in neat films. **h** Nonradiative transition rate ($k_{nr}$) of them in neat films.

heavy-atom effects can significantly enhance the (R)ISC processes due to the abundant spatial interactions between bromine atoms and π systems in the aggregate state (Fig. 3f, and Supplementary Fig. 15). To accurately ascertain the changes in the photophysical processes, we calculated the radiative transition rate ($k_r$), and nonradiative transition rate ($k_{nr}$) of the two HLCT scintillators (Fig. 3g, h). According to the formulas[40] $k_r = \Phi_f/\tau$ and $k_{nr} = 1/\tau - k_r$ (where $\Phi_f$ represents the PLQY and $\tau$ is the PL decay lifetime), the $k_r$ of $1.17 \times 10^8 \, s^{-1}$ in BTD-FLBr largely increases to $2.67 \times 10^8 \, s^{-1}$ in BTD-HeBr, which indicates that a short fluorescence lifetime and a high PLQY value can greatly facilitate the singlet radiative transition process. On the other hand, the higher $k_{nr}$ of BTD-FLBr ($1.06 \times 10^8 \, s^{-1}$) can be attributed to the severe nonradiative transitions via bromine-induced energy dissipation pathways. In contrast, the even lower $k_{nr}$ of BTD-HeBr ($3.80 \times 10^5 \, s^{-1}$) indicates that decoupled π-bromine interactions can significantly suppress non-radiative transitions, which is consistent with the theoretically calculated results. It is found that BTD-HeBr with through-space bromine atoms not only largely improves the radiative rate, but also dramatically suppresses the nonradiative decay, which is beneficial for achieving high-efficiency X-ray imaging.

## Radioluminescence properties

The superior photophysical properties of BTD-HeBr (100% PLQY, short decay time, low $k_{nr}$) lay the foundation for its excellent RL performance. X-ray absorption measurements of BTD-FLBr and BTD-HeBr are performed to investigate their X-ray absorption ability (Fig. 4a). The absorption coefficients of BTD-HeBr ($Z_{max} = 35$, $K_\alpha = 13.5$ keV) was slightly higher than that of BTD-FLBr, which indicates that the through-space bromine atoms exhibit better X-ray photon absorption ability. Moreover, the attenuation efficiency (Supplementary Fig. 25) of BTD-FLBr and BTD-HeBr was calculated to be 69.3 and 75.1% at a thickness of 10 mm, respectively. In a further set of experiments, we explored the RL behaviors of the HLCT scintillators in the solid state (Fig. 4b). Under the same X-ray irradiation (dose rate, 2.023 mGy s$^{-1}$), BTD-FLBr display weak X-ray emission, which is not obviously observable by the naked eye owing to their weak X-ray absorption or low PLQY[41]. In contrast, BTD-HeBr exhibits clearly brighter yellow X-ray luminescence with a narrow FWHM of 56 nm under a continuous X-ray source, which is considerably narrower than that of commercial inorganic scintillators such as CsI:Tl (~150 nm), (Lu$_x$Y$_{1-x}$)$_2$SiO$_5$:Ce (LYSO:Ce) (~80 nm), and Bi$_4$Ge$_3$O$_{12}$ (BGO) (~170 nm). This narrow FWHM can be attributed to the LE character of the HLCT state, where LE's localized π → π* transitions minimize electron density redistribution. Such localization weakens electron-vibronic coupling and environmental perturbation sensitivity, restricting transition energy distribution to yield a compressed spectral width. Additionally, a pulse-height spectrum method shows that BTD-HeBr exhibited a decent light yield of ~13,400 ± 1000 photons MeV$^{-1}$ (Supplementary Fig. 26). The high relative light yield of BTD-HeBr is mainly due to their high X-ray absorption cross-section, unit exciton utilization efficiency and low non-radiative decay. In

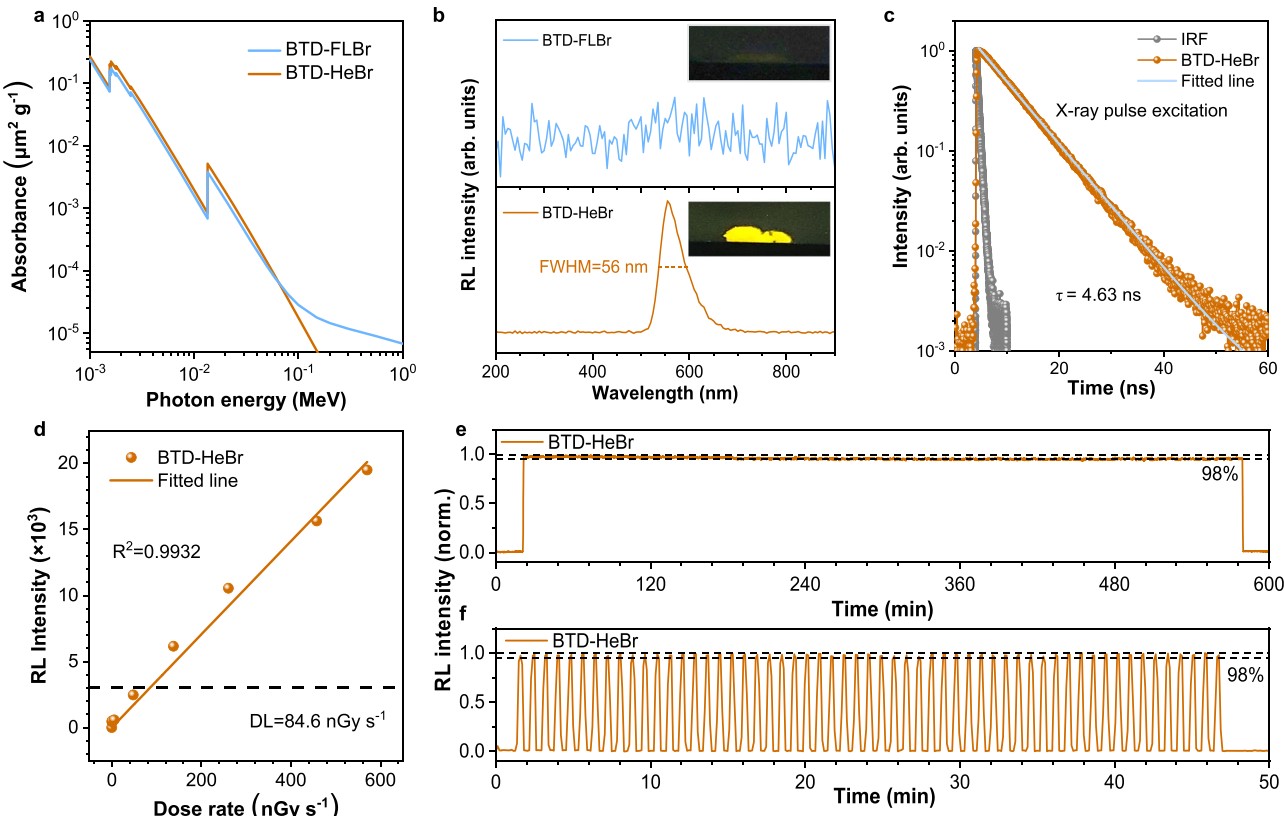

**Fig. 4 | Radioluminescence (RL) performance characterizations of the coupled and decoupled Br-π HLCT scintillators. a** X-ray absorption spectra of BTD-FLBr and BTD-HeBr scintillators measured as a function of X-ray energy[41]. **b** RL spectra of BTD-FLBr and BTD-HeBr taken under X-ray irradiation and their images (size: 1.8 × 0.7 cm, thickness: ~200 μm, X-ray tube voltage: 50 kV, dose rate: 2.023 mGy s$^{-1}$). **c** Luminescence decay curve of BTD-HeBr with X-ray pulse excitation. IRF, impulse response function. **d** The linear relation between X-ray dose rate and RL intensity and detection limit of BTD-HeBr. **e** The RL emission photostability for the BTD-HeBr versus continuous irradiation and repeated on–off cycles (**f**) of X-rays at a dose rate of 2.023 mGy s$^{-1}$.

addition, the luminescence decay time of BTD-HeBr (excited using a transient X-ray) is fitted as 4.63 ns with single exponential decays (Fig. 4c), which is consistent with the PL decay time (3.74 ns). This short lifetime, enabling high temporal resolution, renders it well-suited for rapid dynamic X-ray imaging. Furthermore, the RL intensity of BTD-HeBr was linear response to the X-ray dose rate in the range 0.401–570.1 μGy s$^{-1}$ (Fig. 4d, and Supplementary Fig. 27). Additionally, the detection limit is a measure of the response linearity at low excitation levels and can be defined as the dose rate at which the signal-to-noise ratio equals 3. The detection limit was 84.6 nGy s$^{-1}$ for BTD-HeBr films, which is significantly below the minimum X-ray dose rate requirement for medical X-ray imaging (5.5 μGy s$^{-1}$)[42], demonstrating the highly potential of medical radiography applications. Moreover, BTD-HeBr films exhibit excellent photostability (Fig. 4e). Specifically, when BTD-HeBr was exposed to a high dose rate of X-ray (2.023 mGy s$^{-1}$) for a continuous 600 min, the RL intensity remains at around 98% of the initial value, which is comparable to that of commercial plastic scintillators[2]. Moreover, the emission intensity of BTD-HeBr remains >98% under repeated X-ray excitation for 50 min (57 on–off circles), highlighting its superior photostability.

To achieve high performance and good processability of the HLCT scintillation films for X-ray imaging, glassy neat films are fabricated by melting the HLCT scintillators at a relatively low temperature of 80 °C for X-ray correlated measurements due to the low glass transition temperatures ($T_g$ = 13.4 °C) (Supplementary Fig. 28). The thickness-dependent RL of BTD-HeBr exhibits significantly high intensity with an optimal thickness at 180 μm (Supplementary Fig. 29), and its glassy film shows over 93% transmittance in the 500–800 nm

range—encompassing the entire emission spectrum (Supplementary Fig. 30). This feature can enhance the photon collection efficiency during optical characterization and may realize high resolution for X-ray imaging. The ultra-high X-ray imaging resolution of 49.6 lp mm$^{-1}$ was achieved at a modulation transfer function (MTF) of 0.2 for BTD-HeBr scintillation screen, based on the MTF analysis of standard X-ray edge images (Fig. 5a). To more directly observe and verify the resolution boundary, X-ray imaging was performed using a microresolution test chart (JIMA RT RC-05B, 3–50 μm). As shown in Fig. 5b, c, a spatial resolution of up to 10 μm (i.e., 50.0 lp mm$^{-1}$) can be clearly resolved in a microresolution chart from the gray value intensity along these lines for BTD-HeBr scintillator. A bright-field photograph was shown in Supplementary Fig. 31. As far as we know, such a high imaging resolution exceeds those of all reported organic scintillators (Supplementary Table 2), which supports the high practical X-ray imaging potential of BTD-HeBr scintillators. This resolution is the integrated performance of the entire imaging system (consisting of the scintillator material, optical coupling components, and geometric magnification module), rather than a standalone property of the scintillator itself. Furthermore, we performed imaging tests using BTD-HeBr scintillators to demonstrate their practical value. The BTD-HeBr scintillator film exhibits minimal afterglow, an intrinsic advantage rooted in its fast decay time—this property is critical for eliminating signal persistence in rapid dynamic X-ray imaging. We captured video of fan blades rotating at 400 r min$^{-1}$ using a camera, which enabled clear observation of the holes on the blades without any trailing artifacts. Through analysis of four consecutive frames (each frame spaced ~260 μs), subtle deviations in the fan blades and holes were clearly

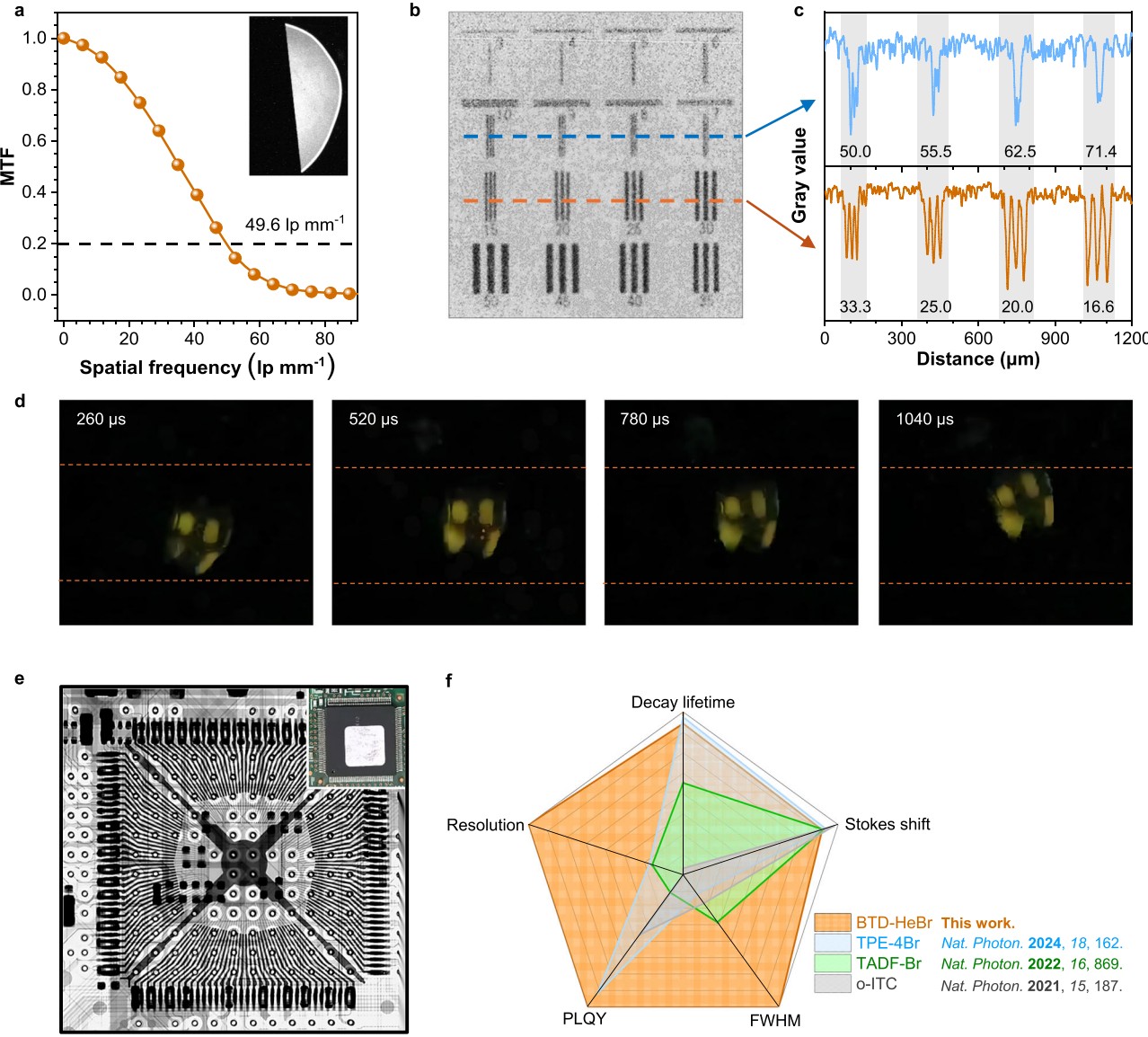

**Fig. 5 | Highly resolved X-ray imaging based on the BTD-HeBr scintillators.**
**a** Modulation transfer function (MTF) of an X-ray slanted-edge image (inset) reveals a spatial resolution of 49.6 lp mm⁻¹ for BTD-HeBr screen. **b** The X-ray image of a microresolution chart (X-ray tube voltage: 50 kV, dose rate: 2.023 mGy s⁻¹). **c** The gray value profiles along the cyan and green lines extracted from the X-ray image of the microresolution chart. Note that the numbers (3–50, unit: μm) in **b** represent the corresponding spatial resolution, which is converted to the numbers in **c** (16.6–71.4) and expressed in lp mm⁻¹. **d** X-ray imaging of rotating blades under high-speed of 3840 fps (size: 3.0 × 3.0 cm). **e** Bright- and darkfield photographs of an electronic chip before and after X-ray exposure (size: 2.5 × 2.5 cm, thickness: ~200 μm, X-ray tube voltage: 50 kV, dose rate: 2.023 mGy s⁻¹). **f** Comparison of typical RTP (TPE-4Br), TADF (TADF-Br), hot exciton (o-ITC) scintillators vs. BTD-HeBr across five dimensions (decay lifetime, Stokes shift, FWHM, PLQY, resolution).

distinguishable (Fig. 5d, and Supplementary Fig. 32), demonstrating the good potential of BTD-HeBr scintillator films in high-fidelity temporal resolution and artifact-free detection. Furthermore, X-ray contrast imaging facilitated the examination of the intricate internal structure of an electronic chip, which is typically impenetrable to the naked eye (Fig. 5e). The intricate architecture of the electronic chip was vividly revealed using a BTD-HeBr scintillating screen. These results demonstrate the promising potential of the HLCT scintillators with specially decoupled heavy atom-π interactions in medical radiography. In addition, compared with previously reported typical RTP (TPE-4Br)[7], TADF (TADF-Br)[9], and hot exciton (o-ITC)[10] scintillators with coupled Br-π interactions, the decoupled Br-π HLCT scintillator BTD-HeBr exhibits the best comprehensive X-ray imaging performance in five dimensions (decay lifetime, Stokes shift, FWHM, PLQY, and spatial resolution), indicating that modulating charge transfer properties and engineering heavy atoms constitute a feasible approach to designing high-performance X-ray scintillators.

## Discussion

In this work, we have developed a promising strategy of organic scintillators to achieve excellent comprehensive performance of RL. Modulating donor-acceptor structures to form HLCT states (LE character for narrow FWHM, CT character for large Stokes shift) and introducing decoupled Br-π interactions (suppressing nonradiative decay for 100% PLQY, enhancing SOC for fast decay) enables the simultaneous achievement of a fast radiative lifetime (3.74 ns), a narrow FWHM (56 nm), a large Stokes shift (110 nm), and a high PLQY (100%). Additionally, we introduced nonconjugated alkyl bromines based on decoupled heavy atom-π interactions for chemically modifying HLCT scintillators. This concept has the following advantages.

First, emitters and heavy atoms interact via spatial interactions to enhance X-ray absorption cross-section. Second, their spatial separation promotes singlet-triplet SOC, facilitating the (R)ISC process and boosting exciton utilization. Third, the nonconjugated architecture avoids heavy atom-induced strong nonradiative decay of singlet excitons. Thanks to the numerous spatial interactions between bromine atoms and π-electrons, BTD-HeBr shows high RISC rate ($2.63 \times 10^8 s^{-1}$), small nonradiative decay ($3.8 \times 10^5 s^{-1}$) and X-ray absorption cross-section in the aggregate state. The decoupled Br-π HLCT scintillators exhibit highly improved scintillation performance (narrow FWHM of 56 nm and detection limit of $84.6 nGy s^{-1}$). In addition, a transparent screen can be obtained by melting glassy films of BTD-HeBr to obtain attractive radioluminescence performance. We believe that the present findings will serve as a benchmark for the fabrication of efficient organic scintillators in commercial applications, including applications in medical radiography and security screening, offering an alternative to high-cost metal complexes and perovskites[43–46].

## Methods

### Materials
Unless otherwise noted, all the chemicals and reagents were obtained from commercial sources (J&K, TCI, Meryer or Sigma Aldrich). The solvents for reactions were distilled and degassed before use. All reactions were carried out in an $N_2$ atmosphere with a dried Schlenk glassware or tube. $^1H$ and $^{13}C$ nuclear magnetic resonance spectra were measured using deuterated $CDCl_3$ as solvent on a Bruker AV 500 spectrometer or a Bruker AVIII 400 spectrometer at room temperature. High-resolution mass spectra were obtained on a GCT premier CAB048 mass spectrometer.

### Characterization
UV-Vis absorption spectra were measured on a PerkinElmer Lambda 950 spectrophotometer. Photoluminescence (PL) spectra, transient PL decay spectra and photoluminescence quantum yields were performed on an Edinburgh FLS1000 fluorescence spectrofluorometer. Thermogravimetric analysis (TGA) was carried out on TA TGA Q5000 under dry nitrogen at a heating rate of 10 °C min⁻¹. Thermal transition was investigated by TA DSC Q1000 under dry nitrogen at a heating rate of 10 °C min⁻¹. Transient absorption spectroscopy is conducted using an Ultrafast Systems Helios femtosecond transient absorption spectrometer. A femtosecond laser amplifier from Light Conversion generates a series of 1030 nm pulses, which are then divided into two separate beams to create the pump and probe pulses. For the probe, the pulses are focused onto a sapphire crystal and a YAG crystal, producing a visible (500–910 nm) and infrared (1100–1600 nm) continuum, respectively. To generate the pump beam centered at 400 nm, an optical parametric amplifier is used. A mechanical delay stage controls the time delay between the pump and probe pulses, ensuring precise measurements.

### Computational details
The ground state geometries were optimized using the DFT method with PBE0-D3 functional at the basis set level of 6-31G (d, p), and the excited-state geometries were optimized using the time-dependent DFT method with PBE0-D3 functional at the basis set level of 6-31G (d, p). The above calculations were performed using Gaussian16 package. The spin-orbital coupling and non-adiabatic coupling constants were calculated based on PBE0/def2-SV(P) by using ORCA 5.0.2. The NTO analyzes were analyzed with Multiwfn.

### Fabrication of BTD-HeBr scintillation films
The scintillation films were prepared using a simple and conventional melt-quenched technique. 500 mg of BTD-HeBr was transferred to a beaker and heated at 120 °C until it melts and the bubbles disappear. Subsequently, the liquid in the beaker mentioned above was poured into a quartz plate, and the scintillation films with a thickness of ~180 microns and a diameter over 2.5 cm were formed through rapidly cooling at room temperature.

### X-ray detection performance and X-ray imaging measurements
The radioluminescence spectra and detection limit of the scintillator was measured using a fiber-coupled fluorescence spectrophotometer (Omni-508 λ 300i, Zolix) equipped with an integrating sphere, and the distance of the X-ray source-to-sample was fixed at 30 cm. During the measurements, the scintillators were closely attached to the circular window of the integrating sphere. The radioluminescence lifetime of BTD-HeBr excited by an X-ray pulse was quantified via time-correlated single photon counting measurements. An ultrashort X-ray pulse (average, 80 ps) was generated by a light-excited X-ray tube (N5084, Hamamatsu). The input excitation light was converted into an electron beam at the photocathode, and then the electron beam was accelerated and focused by the electron lenses to collide with the target, upon which the X-rays were generated and emitted through the beryllium window. For the X-ray imaging measurements, the prepared scintillator film was integrated onto a commercial amorphous silicon TFT photodetector array panel with a pixel size of 85 μm (iRay, NDT0505J, without scintillator) to obtain the X-ray imaging module. During the imaging process, the imaging object was placed on a three-dimensional scanning stage between the X-ray source and the imaging module. To achieve high-resolution X-ray imaging, the relative distances between the X-ray source, the scintillator imaging module, and the object for imaging could be adjusted, thereby enabling image magnification. Additionally, the acquisition program performed electronic amplification processing on the X-ray images collected. The entire imaging system was placed in a lead darkroom box for radiation shielding. The observable X-ray imaging spatial resolution of the samples was determined using a micro-resolution chart (JIMA RT RC-05B, 3–50 μm). In the flash X-ray imaging configuration, the system utilized an X-ray tube (Moxtek Miniature '12 W') as the radiation source. An organic scintillator film was mounted above a reflective prism, which guided the scintillation light toward a CMOS camera (FL-20BW, Tucsen Photonics Co., Ltd). High-speed imaging was acquired using a Huawei Mate 40 Pro device, operating at a frame rate of 3840 fps, under an X-ray dose rate of 703 μGyair s⁻¹ and an exposure time of 500 ms, to record video of a fan rotating at 400 r min⁻¹. The MTF curve was characterized using the slanted-edge method. A sharp-edged X-ray projection was generated by placing a 0.5 mm-thick tungsten sheet on the scintillator at a moderate vertical inclination, after which a region of interest centered on the tungsten edge was selected from the image. The edge spread function (ESF) was derived from this edge profile, and the line spread function (LSF) was obtained via differentiation of the ESF. The MTF was then computed from the Fourier transform ($\mathscr{F}$) of the LSF, as described by the following equation:

$$MTF(\nu) = \mathscr{F}(LSF(x)) = \mathscr{F}\left(\frac{dESF(x)}{dx}\right) \quad (3)$$

where $\nu$ represents the spatial frequency and $x$ denotes the pixel position. All calculations were performed using the SE_MTF_2xNyquis plugin implemented in ImageJ software. The pixel dimension and optical magnification of the imaging system were calibrated by comparing the physical size of the test object with the corresponding pixel count in the recorded image. Every measurement was carried out in ambient air at room temperature.

## Data availability
All data are available in the main text and supplementary materials. The data that support the findings of this study are available from the corresponding authors on request.

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

## Acknowledgements

This work was supported by the National Natural Science Foundation of China (22505108 (C.L.), 52473309 (Y.C.L.), W2412114 (B.X.), 22279059 (B.X.), 52333007 (B.T.Z.), 52273197 (Z.Z.) and 22222905 (P.C.)), Natural Science Foundation of Jiangsu Province (BK20251433 (C.L.) and BK20240083 (B.X.)), and the Fundamental Research Funds for the Central Universities (No. 30925010203 (C.L.) and 30925020113 (B.X.)). This work was also supported by the National Key Research and Development Program of China (2023YFB3810001) (B.Z.T.); the Research Grants Council of Hong Kong (16307020 (B.Z.T.), C6014-20W (B.Z.T.), 27200822 (P.C.)); the Innovation and Technology Commission (ITCC-NERC14SC01) (B.Z.T.); and the Shenzhen Key Laboratory of Functional Aggregate Materials (ZDSYS 20211021111400001) (B.Z.T.); the Science, Technology and Innovation Commission of Shenzhen Municipality (KQTD 20210811090142053) (B.Z.T.). This work was also supported by the National University Research Fund (GK202309001, GK202201015, GK202309022) (Y.C.L.), the Key Research and Development Program in Shaanxi Province of China (2023-YBGY-424) (Y.C.L.), Environment and Conservation Fund (111/2022) (P.C.).

## Author contributions

C.L. designed the experiments and performed the synthesis, major theoretical calculations, and photophysical measurements and wrote the paper. B.X., Y.C.L., and B.Z.T. supervised the project. Y.H.L., B.J., and Z.W. performed X-ray imaging and detection measurements. M.W. performed partial photophysical measurements. F.K. performed transient absorption spectra measurements. Z.L. performed the partly theoretical calculations. X.W. performed partial synthesis. X.L., P.C., Z.Z., R.K., J.L., and S.L. took part in the discussion and gave important suggestions. All authors approved the final version of the manuscript.

## Competing interests

The authors declare no competing interests.
