## [Transparent Peer Review file · Nature Communications]

High-Resolution X-Ray Imaging via Spatially Decoupled Heavy-Atom Antennas in Organic Scintillators

Corresponding Author: Professor Ben Tang

Version 0:

Reviewer comments:

Reviewer #1

(Remarks to the Author)

In most responses, I understand and am convinced by explanations. However, I still cannot miss the calculation way of light yield.

Regarding to my last comment 2, unfortunately, your understanding is wrong. Your data (and data in reference Nature paper) are all RL spectrum, and they cannot be used to determine the absolute scintillation light yield. High impact papers are not an evidence that the methodology is correct, and in terms of methodology to determine the light yield, all these Nature paper are wrong. For instance, if the stopping power can be calibrated (I have a strong suspicion on this point since there is no calculation tool especially for low energy range, and also, there is no calculation tool to consider non-proportional response of scintillators), then, how do you correct afterglow and intrinsic radioactivity? Since the light yield is one of the most important properties for scintillators, I cannot miss an obvious mistake.

To conduct a pulse height measurement, you can measure by using ^{241}Am 60 keV, or other RIs having lower energy.

Reviewer #2

(Remarks to the Author)

The authors have reorganized the manuscript and addressed several reviewer concerns. I would like to acknowledge that the overall material design strategy and the authors' approach to modulating the HLCT process are conceptually interesting and scientifically meaningful. These aspects already provide a solid foundation for the work and offer genuine value to the scintillator community.

However, regarding the measurement of light yield, both Reviewer 1 and Reviewer 2 raised fundamental doubts that have not been satisfactorily addressed. Although the authors cite several Nature-series publications to justify their method, I respectfully disagree with this reasoning. Radiation detection is a mature field with decades of methodological development, and many rigorous measurement standards—particularly those published in journals such as Nuclear Instruments and Methods A (NIMA)—have been widely accepted and followed for years. The fact that a few high-impact publications used a non-standard method for material demonstration does not validate the method itself, especially when those works did not aim to establish or benchmark a reliable measurement protocol.

Therefore, I strongly recommend that the authors perform a pulse-height spectrum (PHS) measurement to determine the light yield. From a scientific perspective, the number of photons generated during radiation detection is extremely low, and the resulting Poisson noise is substantial. A single-shot RL spectrum inherently suffers from large statistical fluctuations, whereas PHS provides tens of thousands of detection events, enabling proper statistical averaging and meaningful light-yield determination.

I also hope the authors take this issue seriously in the interest of scientific rigor. If light yield is overestimated due to an

insufficiently robust measurement, it may mislead subsequent materials development efforts. Several authors of this manuscript have long-standing experience in radiation detection and will likely continue contributing to this field; I am confident they would not want a methodological ambiguity in this paper to obscure their future achievements.

Moreover, as noted above, the material design concept and the HLCT modulation strategy are sufficiently innovative. The authors do not need to pursue an unrealistically high light yield to justify the significance of the work. From a practical standpoint, powder samples inherently face limitations in PHS measurements (e.g., internal scattering). In this context, it may be more rigorous to avoid reporting a quantitative light-yield value and instead present a relative comparison—for example, demonstrating that the steady-state RL intensity is several times higher than that of a commercial scintillator under identical laboratory conditions, while explicitly noting that such comparison cannot be directly converted into an absolute light yield. This approach avoids overclaiming while still supporting the material's potential as a scintillator.

Regarding the spatial-resolution metric, since the authors have clarified that the reported 50 lp/mm arises from the geometric magnification of the imaging system, this parameter is not intrinsic to the material. Yet the manuscript repeatedly highlights “resolution” in the Abstract and other key sections, which may mislead readers to assume that the material itself enables this performance. I strongly recommend removing or substantially downplaying these resolution claims, as they are not aligned with the core innovation of the work.

Finally, for the radioluminescence lifetime measurements, I suggest using transient X-ray excitation to improve the signal-to-noise ratio. The current decay curves exhibit intensities too low to allow logarithmic analysis, making it difficult to resolve both fast and slow lifetime components. A more rigorous lifetime characterization is important not only for this manuscript but also for future advances in HLCT-based scintillators.

Version 1:

Reviewer comments:

Reviewer #1

(Remarks to the Author)

Now, the measurement methodology of LY is correctly done, and now I can recommend a publication. Strictly speaking, we must consider the non-proportionality, but it will be beyond the focus of this paper.

Reviewer #2

(Remarks to the Author)

The authors have well addressed all the issues from all reviewers. I have no more questions.

Responds Letter

Reviewer #1 (Remarks to the Author):

In most responses, I understand and am convinced by explanations. However, I still cannot miss the calculation way of light yield.

Regarding to my last comment 2, unfortunately, your understanding is wrong. Your data (and data in reference Nature paper) are all RL spectrum, and they cannot be used to determine the absolute scintillation light yield. High impact papers are not an evidence that the methodology is correct, and in terms of methodology to determine the light yield, all these Nature paper are wrong. For instance, if the stopping power can be calibrated (I have a strong suspicion on this point since there is no calculation tool especially for low energy range, and also, there is no calculation tool to consider non-proportional response of scintillators), then, how do you correct afterglow and intrinsic radioactivity? Since the light yield is one of the most important properties for scintillators, I cannot miss an obvious mistake.

To conduct a pulse height measurement, you can measure by using ^{241}Am 60 keV, or other RIs having lower energy.

Respond: We sincerely appreciate your insightful and critical comments on the determination methodology of absolute scintillation light yield (LY) of scintillators. We fully agree with your viewpoint that RL spectra only reflect the relative luminescent properties of scintillators, and cannot be applied for quantitative calculation of absolute LY. The ambiguous description of the correlation between RL data and absolute LY in the original manuscript has been completely revised. Meanwhile, we abandoned the inappropriate method of deriving LY from RL spectra adopted in some published studies, and employed a low-energy radiation source-based PHA method. As a universally recognized standard approach for quantitative LY measurement of scintillators in the low-energy region, this method can effectively avoid the inherent limitations of the RL spectroscopy method. Per your suggestion, we have supplemented the LY measurements via the pulse height analysis (PHA) method using a ^{241}Am (59.5 keV) radiation source and systematically addressed the methodological flaws in the

original manuscript. The detailed improvements and experimental details are summarized as follows:

The light yield (LY) of the samples was quantified by analyzing the pulse height spectra obtained under excitation with standard ^{241}Am (59.5 keV) γ -ray sources. In actual measurements, we note that the α particles from ^{241}Am decay exhibit weak penetration, which would preclude the installation of a light shield and cause severe photon loss. To eliminate this interference, we employed a black aluminum foil to absorb the α particles, and the data acquisition system was set to collect signals only within the energy window corresponding to 59.5 keV. The electrical signals generated by the photomultiplier tube (PMT) were collected and digitized using a dedicated data acquisition system. Subsequently, the light yield value was derived by determining the position of the full-energy peak in the spectrum and conducting a normalization comparison between this peak and the single-photon response of the PMT system. The value can be calculated using the following relationship:

$$LY = \frac{Bin \times K}{E \times \eta}$$
$$K = 0.043066$$

where Bin is the channel number corresponding to the full-energy peak; K is the system-related correction parameter, including the channel number corresponding to a single photon, amplification factor, single-photon correction, and other items; E is the energy corresponding to the full-energy peak (unit: MeV); and η is the collection efficiency of the photomultiplier tube for the fluorescence band of the samples.

Supplementary Fig. 26. (a) Pulse height spectrum of BTD-HeBr and LYSO:Ce under ^{241}Am excitation. (b) The quantum efficiency of PMT.

As shown in Supplementary Fig. 26, the two samples present a clear photopeak of the pulse height spectra and the channel numbers are 91 and 224, respectively. The emission-weight quantum efficiency (EWQE) of the PMT used in the characterization is 23.10%. For the collection efficiency of the PMT, the average fluorescence efficiency $\eta=10.35\%$ was obtained via convolution calculation of the radioluminescence (RL) spectrum of the sample and the fluorescence collection efficiency of the PMT. Therefore, LYSO:Ce has a light yield of $33\,000 \pm 1500$ photons/ MeV, which is very close to the references confirming the satisfactory accuracy of the gamma-spectrometer and the described method. Since the channel number is linearly proportional to the light yield under the identical test conditions (same K , E , and η), the light yield of BTD-HeBr was derived as approximately $13,400 \pm 600$ photons/MeV. The error bars were determined from the standard deviation of three independent measurements and the uncertainty of PMT quantum efficiency calibration. This value is higher than that of most organic scintillators, such as EJ200 (10000 photons/MeV).

Reviewer #2 (Remarks to the Author):

The authors have reorganized the manuscript and addressed several reviewer concerns.

I would like to acknowledge that the overall material design strategy and the authors' approach to modulating the HLCT process are conceptually interesting and scientifically meaningful. These aspects already provide a solid foundation for the work and offer genuine value to the scintillator community.

Respond: We would like to express our sincere gratitude for your positive feedback and recognition of our work. Your affirmation of the conceptual novelty of the material design strategy and the scientific significance of modulating the hot exciton (HLCT) process is a great encouragement to our research team, and it also strengthens our confidence in the value of this work to the scintillator community.

However, regarding the measurement of light yield, both Reviewer 1 and Reviewer 2 raised fundamental doubts that have not been satisfactorily addressed. Although the authors cite several Nature-series publications to justify their method, I respectfully disagree with this reasoning. Radiation detection is a mature field with decades of methodological development, and many rigorous measurement standards—particularly those published in journals such as Nuclear Instruments and Methods A (NIMA)—have been widely accepted and followed for years. The fact that a few high-impact publications used a non-standard method for material demonstration does not validate the method itself, especially when those works did not aim to establish or benchmark a reliable measurement protocol.

Therefore, I strongly recommend that the authors perform a pulse-height spectrum (PHS) measurement to determine the light yield. From a scientific perspective, the number of photons generated during radiation detection is extremely low, and the resulting Poisson noise is substantial. A single-shot RL spectrum inherently suffers from large statistical fluctuations, whereas PHS provides tens of thousands of detection events, enabling proper statistical averaging and meaningful light-yield determination.

I also hope the authors take this issue seriously in the interest of scientific rigor. If light yield is overestimated due to an insufficiently robust measurement, it may mislead subsequent materials development efforts. Several authors of this manuscript have long-standing experience in radiation detection and will likely continue contributing to

this field; I am confident they would not want a methodological ambiguity in this paper to obscure their future achievements.

Moreover, as noted above, the material design concept and the HLCT modulation strategy are sufficiently innovative. The authors do not need to pursue an unrealistically high light yield to justify the significance of the work. From a practical standpoint, powder samples inherently face limitations in PHS measurements (e.g., internal scattering). In this context, it may be more rigorous to avoid reporting a quantitative light-yield value and instead present a relative comparison—for example, demonstrating that the steady-state RL intensity is several times higher than that of a commercial scintillator under identical laboratory conditions, while explicitly noting that such comparison cannot be directly converted into an absolute light yield. This approach avoids overclaiming while still supporting the material's potential as a scintillator.

Respond: We sincerely appreciate your incisive and constructive comments on the measurement of scintillation light yield (LY), which are pivotal for further improving the methodological rigor and scientific credibility of our work. We fully endorse your viewpoint that rigorous measurement standards established in specialized journals such as Nuclear Instruments and Methods in Physics Research Section A (NIMA) should be the fundamental guideline for LY determination, rather than simply citing methods from high-impact publications that are not dedicated to benchmarking measurement protocols. We deeply appreciate your reminder regarding the potential risk of misleading future materials development efforts due to inaccurate LY data. In response to your strong recommendation, we have abandoned the use of radioluminescence (RL) spectra for LY calculation, and have re-performed the LY measurement following the standard pulse-height spectrum (PHS) method.

The key details of the optimized experiment are summarized as follows:

The light yield (LY) of the samples was quantified by analyzing the pulse height spectra obtained under excitation with standard ^{241}Am (59.5 keV) γ -ray sources. In actual measurements, we note that the α particles from ^{241}Am decay exhibit weak penetration, which would preclude the installation of a light shield and cause severe

photon loss. To eliminate this interference, we employed a black aluminum foil to absorb the α particles, and the data acquisition system was set to collect signals only within the energy window corresponding to 59.5 keV. The electrical signals generated by the photomultiplier tube (PMT) were collected and digitized using a dedicated data acquisition system. Subsequently, the light yield value was derived by determining the position of the full-energy peak in the spectrum and conducting a normalization comparison between this peak and the single-photon response of the PMT system. The value can be calculated using the following relationship:

$$LY = \frac{Bin \times K}{E \times \eta}$$

$$K = 0.043066$$

where Bin is the channel number corresponding to the full-energy peak; K is the system-related correction parameter, including the channel number corresponding to a single photon, amplification factor, single-photon correction, and other items; E is the energy corresponding to the full-energy peak (unit: MeV); and η is the collection efficiency of the photomultiplier tube for the fluorescence band of the samples.

Supplementary Fig. 26. (a) Pulse height spectrum of BTD-HeBr and LYSO:Ce under ^{241}Am excitation. (b) The quantum efficiency of PMT.

As shown in Supplementary Fig. 26, the two samples present a clear photopeak of the pulse height spectra and the channel numbers are 91 and 224, respectively. The emission-weight quantum efficiency (EWQE) of the PMT used in the characterization is 23.10%. For the collection efficiency of the PMT, the average fluorescence efficiency $\eta=10.35\%$ was obtained via convolution calculation of the radioluminescence (RL) spectrum of the sample and the fluorescence collection efficiency of the PMT. Therefore, LYSO:Ce has a light yield of $33\,000 \pm 1500$ photons/ MeV, which is very close to the references confirming the satisfactory accuracy of the gamma-spectrometer and the described method. Since the channel number is linearly proportional to the light yield under the identical test conditions (same K , E , and η), the light yield of BTD-HeBr was derived as approximately $13,400 \pm 600$ photons/MeV. The error bars were determined from the standard deviation of three independent measurements and the uncertainty of PMT quantum efficiency calibration. This value is higher than that of most organic scintillators, such as EJ200 (10000 photons/MeV).

Regarding the spatial-resolution metric, since the authors have clarified that the reported 50 lp/mm arises from the geometric magnification of the imaging system, this parameter is not intrinsic to the material. Yet the manuscript repeatedly highlights “resolution” in the Abstract and other key sections, which may mislead readers to assume that the material itself enables this performance. I strongly recommend removing or substantially downplaying these resolution claims, as they are not aligned with the core innovation of the work.

Respond: We sincerely appreciate your precise and constructive comment on the spatial-resolution metric of our work. Your point about the spatial resolution is entirely accurate, and we apologize for the ambiguity and misleading implications caused by our inappropriate emphasis on this parameter in key sections of the original manuscript. We fully agree with your suggestion that the overemphasis on resolution is inconsistent with the core innovation of this work. In the revised manuscript, we have carried out the following targeted revisions to address this issue:

We have removed all highlighted descriptions of the 50 lp/mm spatial resolution in the Abstract and Conclusion sections—parts that are most likely to draw readers' primary attention. For the remaining mentions in the Results and Discussion section, we have clearly labeled “this resolution as the integrated performance of the entire imaging system (consisting of the scintillator material, optical coupling components, and geometric magnification module), rather than a standalone property of the scintillator itself.”

We have adjusted the narrative logic of the manuscript to center on its core innovation: the rational design strategy of the scintillator material and the effective modulation of the HLCT process. The discussion of imaging resolution is now positioned as a demonstration of practical application potential of the material, rather than a key merit to highlight the innovation of the material itself.

We deeply recognize that rigorous differentiation between material intrinsic properties and system integrated performance is essential to maintaining the scientific integrity of the manuscript and avoiding misleading readers in the scintillator community. The revised content has been carefully checked to ensure that all descriptions related to spatial resolution are accurate and consistent with the core value of this work.

Finally, for the radioluminescence lifetime measurements, I suggest using transient X-ray excitation to improve the signal-to-noise ratio. The current decay curves exhibit intensities too low to allow logarithmic analysis, making it difficult to resolve both fast and slow lifetime components. A more rigorous lifetime characterization is important not only for this manuscript but also for future advances in HLCT-based scintillators.

Respond: We sincerely appreciate your constructive suggestion on optimizing the radioluminescence (RL) lifetime measurement methodology. We fully agree that a rigorous lifetime characterization is critical for both interpreting the HLCT process in this manuscript and guiding the development of HLCT-based scintillators in future studies.

In response to your recommendation, we have completely re-performed the RL lifetime measurements using a transient X-ray excitation source to improve the signal-to-noise ratio (SNR) of the decay curves. The key details of the optimized experiment and data analysis are summarized as follows:

The radioluminescence lifetime of BTD-HeBr excited by an X-ray pulse was quantified via time-correlated single photon counting measurements. An ultrashort X-ray pulse (average, 80 ps) was generated by a light-excited X-ray tube (N5084, Hamamatsu). The input excitation light was converted into an electron beam at the photocathode, and then the electron beam was accelerated and focused by the electron lenses to collide with the target, upon which the X-rays were generated and emitted through the beryllium window.

In addition, the luminescence decay time of BTD-HeBr (excited using a transient X-ray) is fitted as 4.63 ns with single exponential decays (**Fig. 4c**), which is consistent with the PL decay time (3.74 ns).

Once again, we thank you for your insightful guidance. This optimization of the lifetime measurement methodology not only improves the data quality of this work but also establishes a standardized protocol for our future research on HLCT-based scintillators.